# Voltage-Gated Proton Channels in the Tree of Life

**DOI:** 10.3390/biom13071035

**Published:** 2023-06-24

**Authors:** Gustavo Chaves, Christophe Jardin, Christian Derst, Boris Musset

**Affiliations:** 1Center of Physiology, Pathophysiology and Biophysics, The Nuremberg Location, Paracelsus Medical University, 90419 Nuremberg, Germany; gustavo.chaves@pmu.ac.at (G.C.); christophe.jardin@pmu.ac.at (C.J.); christian.derst@pmu.ac.at (C.D.); 2Center of Physiology, Pathophysiology and Biophysics, The Salzburg Location, Paracelsus Medical University, 5020 Salzburg, Austria

**Keywords:** Aplysia, insect, mollusks, Lophotrochozoa, Ecdysozoa, coccolithophores, voltage-gated proton channel, NADPH oxidase, voltage-sensing, pH-dependent gating

## Abstract

With a single gene encoding H_V_1 channel, proton channel diversity is particularly low in mammals compared to other members of the superfamily of voltage-gated ion channels. Nonetheless, mammalian H_V_1 channels are expressed in many different tissues and cell types where they exert various functions. In the first part of this review, we regard novel aspects of the functional expression of H_V_1 channels in mammals by differentially comparing their involvement in (1) close conjunction with the NADPH oxidase complex responsible for the respiratory burst of phagocytes, and (2) in respiratory burst independent functions such as pH homeostasis or acid extrusion. In the second part, we dissect expression of H_V_ channels within the eukaryotic tree of life, revealing the immense diversity of the channel in other phylae, such as mollusks or dinoflagellates, where several genes encoding H_V_ channels can be found within a single species. In the last part, a comprehensive overview of the biophysical properties of a set of twenty different H_V_ channels characterized electrophysiologically, from Mammalia to unicellular protists, is given.

## 1. Proton Channels in Mammals

Voltage-gated proton currents were measured for the first time in eukaryotes using voltage-clamp. The investigation of an ion channel that exclusively conducts a subatomic particle, the proton, originates from a mollusk, the snail *Helix aspersa* [1]. Before its actual discovery in a living organism, the necessity of a functional voltage-gated proton-selective channel (H_V_) had been anticipated. Fogel and Hastings [2] predicted the voltage-gated proton channel to be the trigger for the bioluminescence reaction dinoflagellates, one of the first eukaryotes spawned in evolution. The second prediction had far-reaching consequences for human health. Henderson et al. [3] prognosticated the voltage-gated proton channel to be essential for the respiratory burst of human phagocytes. The respiratory burst is a greatly increased oxygen consumption of a cell where molecular oxygen is reduced to superoxide [4], and it is essential to kill pathogens. Yet, the first recording of H_V_ in mammals was performed in respiratory tissue (alveolar type 2 cells) of rats by DeCoursey [5]. Type 2 cells, as far as we know, do not perform a respiratory burst. Shortly afterwards, the detection in human tissue was achieved by three groups simultaneously [6,7,8]. Here, two innate immune cells were investigated, freshly isolated human neutrophils and the HL-60 cell line representing human granulocytes. Bernheim et al., 1993, found the channel in human skeletal muscle tissue, which undoubtedly is a strongly metabolically active tissue not performing the respiratory burst. An intensive investigation on the connection between H_V_ and the respiratory burst and the role of the channel in cell types lacking immunological functions in human and mammalian tissue followed. At present, there is no indication of H_V_ channels in prokaryotes. In this first part of the review, we categorize the expression of voltage-gated proton channels in two sections. Firstly, cell types in mammals expressing H_V_ capable of performing the respiratory burst, and secondly, mammalian tissue expressing H_V_ although incapable of conducting a respiratory burst.

### 1.1. What Are the Pillars of the Respiratory Burst?

A prerequisite key element of the respiratory burst is the NADPH oxidase, a multiprotein complex that oxidizes NADPH and reduces molecular oxygen to superoxide or hyperoxide (O_2_^−^). The functioning of the enzyme complex depends on assembly of its components: gp91, p47, p67, p40, p22, and Rac. The function of translocating electrons resides in the transmembrane heterodimer gp91 and p22 (NOX2), where the gp91 subunit harbors all parts needed for electron transfer including two heme groups. Of the four cytosolic components, Rac and p67 are essential to activate NOX2. The subunit p47 appears inessential for activation in some cell-free systems, similar to p40 [9]. As the name obviously states, NADPH is oxidized, and electrons are moved through gp91 across the membrane. This translocation of charge across the isolator plasma membrane generates an electrical potential. The reversal potential of the electrons depends on the midpoint potential of the two redox pairs NADPH/NADP^+^ + H^+^ (−320 mV), and the midpoint potential of O_2_ to O_2_^−^ (−160 mV). Furthermore, it is subjected to the concentration of the charged particles. Ergo, depolarizations of the membrane up to +200 mV are potentially reachable. The human NADPH oxidase in eosinophils and polymorphonuclear leucocytes (PMN) is voltage-dependent [10], ceasing electron flux at high depolarizations. Therefore, charge compensation during the respiratory burst is fundamental, preventing the sudden end of the immune response due to massive depolarization. The second pilar of the respiratory burst is the voltage-gated proton channel, the main focus of this review.

H_V_1 is the most complementary protein of NADPH oxidase during the respiratory burst. H_V_1 opens by depolarization and is controlled by the proton gradient across the membrane. The NADPH oxidation during the respiratory burst accumulates protons inside the cytosol that decrease the intracellular pH (pH_i_) (Figure 1). The change of the pH outside the cell (pH_o_) is negligible due to the immense volume of the extracellular space. Accordingly, the pH gradient (ΔpH) increases and favors H_V_1 opening. Moreover, the outward conduction of protons repolarizes the cell membrane. Consequently, the membrane potential is regulated by the equilibrium between the electron and the proton outflux, avoiding the reduction in the translocation of electrons due to massive membrane depolarizations. Theoretically, the system is then running endlessly as long as there is supply of NADPH. In conclusion, two independent proteins, originated from distinct genes, work synergistically together in one of the most important immune responses of the human body. 

### 1.2. Which Other Physiological Functions Are Affected by H_V_1? 

Foremost, the cell’s pH homeostasis is affected. Proton channels prevent acidosis of the cytosol by conducting the protons out of the cell [7]. The conduction is driven by the electrochemical gradient without any additional requirement of energy [11]. Securing normal cellular pH prevents the cell from all the negative effects of proton accumulation [12]. 

Interestingly, H_V_ additionally sustains Ca^2+^ influx due to store operated Ca^2+^ channels. The concept is comparable to the one found in the potassium channel functioning, especially in Ca^2+^ activated K^+^ channels. The Ca^2+^ influx depolarizes the cell membrane, while the efflux of H^+^ repolarizes it, balancing the charge [13]. Moreover, H_V_ channels appear to be necessary for cell migration [13,14]. They play a role in the exocytosis machinery and granula release, both essential to the immune system [15,16]. In mouse eosinophils, they prevent cell death, shown by H_V_1-KO eosinophils’ decreased survivability [17]. Proton channels support the survival of cancer cells, most unwanted by the host [18]. Invasiveness of metastatic cancer is supported by the function of H_V_1 channels, and the channel itself is evidently a potential biomarker for metastatic cancer [19,20,21]. In microglia, the expression of H_V_1 regulates polarization towards an M1 proinflammatory and an M2 anti-inflammatory cell subset [22]. One of the most intriguing topics is their contribution to metabolic function. In pancreatic β-cells, Kupffer cells, and in adipocytes, they are involved in hormone release, glucose metabolism, and control of glucotoxicity [23,24,25]. In the immune system, B-cells and B-cell receptors are affected by proton channels [26,27], as well as dendritic cells where proton channels are involved in pattern recognition [28]. H_V_1 knock out causes T-cell-related autoimmune diseases [29]. H_V_1 is strongly expressed in lung tissue, and the function there ranges from acidification of the airways to CO_2_ homeostasis [30,31]. The channel is essential to sperm maturation [32,33] but also functional in human oocytes [34], affecting reproduction. Myeloid-derived suppressor cells are suggested to control the adaptive immune system via H_V_1 function [35]. 

A list of tissue expressing H_V_1 and its functions there is provided in Table 1. The investigation of tissue expression and the function of voltage-gated proton channels has just begun. Potential major breakthroughs are expected in future, especially in this thriving medically relevant field. 

## 2. Voltage-Gated Proton Channels in Evolution

As a member of the superfamily of voltage-gated ion channels, proton channels share the membrane spanning domain including the voltage-sensor in S4 with other members of this superfamily, such as voltage-gated potassium (K_V_), sodium (Na_V_), and calcium channels (Ca_V_), TRP channels, hyperpolarization-activated HCN channels, and cyclic-nucleotide-activated CNG channels [107]. Whereas the basic structure of these channels comprises six transmembrane segments, proton channels have lost the typical S5–S6 pore forming domain of other voltage-gated channels, establishing a new H^+^-selective permeation pathway within the first four transmembrane segments [108] (Figure 2). 

Among single mammalian species, the diversity of voltage-gated ion channel is huge. For Na_V_ and Ca_V_ channels, ten different genes have been found. Each Na_V_ and Ca_V_ channel is formed by four domains of the voltage-gated superfamily (S1–S6), all located on one polypeptide chain and encoded by one single gene. Within the K_V_-family, up to 40 different genes have been described. Although the K_V_ channels are also formed by four voltage-gated domains, only one S1–S6 domain is encoded per gene, opening the possibility for the formation of homo- or hetero-tetrameric ion channels. For the exploration of biophysical properties and pharmacological differences among these ion channels, the enormous diversity was of great benefit. Differences in channel function could easily be predicted from simple comparison of the primary amino acid sequence using multiple alignments with functional properties of the analyzed channels. Site-directed mutagenesis approaches, construction of chimeric ion channels, or the analysis of the formation of heteromeric channels (in K_V_) finally gave experimental proof of the investigated structure–function relationship.

For voltage-gated proton channels the situation in mammals is completely different as only one single proton channel (H_V_1) is encoded by one single gene (HVCN1). In humans, this gene is found on chromosome 12q24.11, and seven exons are encoding H_V_1 [111]. Although the existence of two different splice variants in humans has been described [27], the diversity of voltage-gated proton channels in mammals is extremely low, excluding the possibility of sequence comparison and biophysical and pharmacological analysis by “sequence exchange experiments” using paralogous channels. Therefore, the identification and functional analysis of more distantly related voltage-gated proton channels is of great importance. As described in the next chapter, H_V_ channels are found in many eukaryotes from unicellular organisms to most animal species.

### 2.1. The H_V_ Channel “Tree of Life”

Since the 1990’s, it is well established that protein structures typical for potassium, sodium, or calcium channels have already evolved in prokaryotes [112,113,114,115,116]. As a central structural feature, the S1–S4 transmembrane region, harboring the voltage-sensor and dominating the structure of eukaryotic H_V_1 channels, is found in many prokaryotic voltage-gated ion channels. However, as known so far (and verified by database searches), all these prokaryotic voltage-sensing domains are accompanied by a potassium, sodium, or calcium selective pore domain (S5–S6). There is no evidence of typical H_V_1-type proton channels in eubacteria and archaea. In addition, the totally conserved tryptophan in S4 of eukaryotic H_V_1 channels is not found in prokaryotic S1–S4 segments. H_V_ channels found in protists, representing the “early H_V_ evolution”, show significant identity (28–36%) and similarity (~50–60%) to eukaryotic voltage-gated sodium and calcium channels (Table 2). Hence, it can be concluded that H_V_ channels have evolved in an early eukaryote by a split of a region encoding S1–S4 from a eukaryotic Na_V_ or Ca_V_ gene. This hypothesis is supported functionally and structurally by mutations within voltage-sensing elements (arginine residues) of sodium channels rendering these channels permeable for cations including protons through this gating-pore element [117,118].

The appearance of H_V_ channels within the phylogenetic tree of life is shown in Figure 3. A list summarizing the identified H_V_ channel proteins and their GenBank accession numbers is given in Appendix A. For green-labeled clades, the presence of H_V_1 channels and their genes is supported by experimental data or database entries. Unlabeled clades in Figure 3 may either not possess H_V_ channels or the respective data volume coverage may be yet too small to identify a proton channel. Many species within this tree of life lack any typical H_V_ channel. This is exemplified in Table 3 where the presence of an H_V_ channel and its respective gene was analyzed in selected and well-characterized model systems. In many scientific standard model systems, such as the fly *Drosophila melanogaster*, the yeast *Saccharomyces cerevisiae*, the nematode worm *Caenorhabditis elegans*, and the slime mold *Dictyostelium discoideum*, absolutely no indication for an expressed H_V_ channel was found, nor was an H_V_ channel gene found in the respective genomes. In these cases, H_V_ channels might not be essential for life and many species survive without it. On the other hand, a majority of the eukaryotic species analyzed possess at least one H_V_ channel gene and, for some of them, even several distinct H_V_ channel genes have been identified. This, in turn, shows that H_V_ channel genes may be of significant evolutionary advantage and that H_V_ channels may have gathered many novel functions during evolution yet to be discovered and analyzed.

#### 2.1.1. H_V_ Channels in Chordata

H_V_1 channels genes were found initially in mammals. In 2006, Ramsey et al. and Sasaki et al. identified and characterized the first H_V_1 sequences in human and mice, respectively. Among the first publications, a proton channel homolog in the more distantly related invertebrate tunicate *Ciona intestinalis* was described [119], showing 46% identity to mammalian H_V_1 channels, and indicating for the first time, that H_V_ channels are present in non-mammalian animals as well. In the following years, H_V_1 channels were also found within other vertebrates, such as zebra fish [120].

Curiously, in the clawed frog *Xenopus laevis*, a recent gene duplication gave rise to two independent genes (NC_054371 and NC_054372) located on separate loci of chromosome 1 (1L and 1S, respectively). As the encoded H_V_ channels (XM_018249100 and XM_018244209) have 90% of identity and 94% of similarity (Appendix A), future experiments regarding cellular expression and biophysical properties will show if they have already evolved different functions. 

Besides tunicates, Cephalochordata represents an evolutionary old group of chordates with the lancelet as the main species. Sequence analysis of *Branchiostoma belcheri* unambiguously identified three independent channels encoded by three different genes (Appendix A). Whereas one *Branchiostoma* channel is a typical H_V_ channel (XR_002139895), the other two channels incorporated in their structures either a large N-terminal domain (XM_019760615) or a large C-terminal domain (XM_019764911) of unknown function.

#### 2.1.2. H_V_ Channels in Ecdysozoa

Ecdysozoa is a monophyletic superphylum of protostome animals characterized by their ability to grow by ecdysis and include, for example, Arthropoda, Nematoda, and Priapulida. As the largest group within Arthropoda, insects were long thought to lack a proton channel. Indeed, none of the early sequenced dipteran genomes of *Drosophila*, *Anopheles*, or *Aedes* contained a gene encoding a proton channel. In 2016, H_V_1 channels were however found in evolutionary “old” basal hexapodes (Zygentoma, Archeognatha, Diplura, and Protura), and one H_V_ channel from the silver fish *Nicoletia phytophila* was functionally characterized [121]. Later, in the polyneopteran stick insects *Extatosoma tiaratum*, another H_V_ channel was analyzed [122]. Whereas both channels share all typical structural and functional features of proton channels (see below), the *Extatosoma* H_V_ channel was the first proton channel to be characterized harboring a glutamate residue within the S1 segment instead of the usual aspartate. A careful analysis of expression and genomic databases indeed showed that H_V_ channel genes are absent from genomes of hemipteran or holometabolan insects, including all Diptera. Therefore, “modern” insects have lost their H_V_ gene during evolution [122].

In the second major group of arthropodes, the Chelicerata, several species contain more than one H_V_ gene. The tick *Ixodes scapularis*, the varroa mite *Varroa destructor*, and the common house spider *Parasteatoda tepidariorum* contain two different channels (Appendix A). In the Atlantic horseshoe crab *Limulus polyphemus,* even three distinct H_V_ channels and their respective genes could be identified in TSA expression and in genome databases, respectively (Appendix A). The first identified *Limulus* H_V_ channel is a typical, rather small (<250 amino acids) H_V_ channel and has only ~30% identity with the other two closely related (~75% identity) and much larger (>450 amino acids) *Limulus* H_V_ channels. As in the insect *Extatosoma*, the large *Limulus* H_V_ channels possess a glutamate residue within the S1 selectivity filter.

In crustaceans, one typical H_V_ channel is regularly observed, whereas many Myriapoda seem to possess two different H_V_ channels as exemplified by *Scolopocryptops rubiginosus* (Appendix A). Finally, in the only Priapalida species analyzed (*Priapulus caudatus*), a single typical H_V_ channel was identified.

The situation for the second major Ecdysozoa phylum, Nematoda, is slightly different as sporadic H_V_1 homologs were only identified in few Enoplea, including the pathogene *Trichinella spiralis* and three Chromadorea species. For most other Nematoda, including the model system *Caenorhabditis elegans*, no H_V_1 gene was found.

#### 2.1.3. H_V_ Channels in Other Metazoa

A closer look at GenBank TSA and genomic databases of other metazoan phylae unambiguously identified a single H_V_1-like proton channel also present in Hemichordata (*Saccoglossus kowalevskii*). In Echinodermata, a single H_V_ channel was already described and functionally characterized in the sea urchin *Strongylocentrotus purpuratus* [123].

In Platyhelminthes, two different H_V_ channels were identified in some species. For instance, in the bilharzia causing parasitic flatworms (*Schistosoma*), two related channels (~26% identity, ~56% similarity) share an unusual structural feature of an elongated S3-S4 loop. In addition, in the genome of *Schistosoma haematobium*, both genes are located on the same scaffold (NW_023366479) separated by only ~270,000 bp, indicating a gene duplication after Platyhelminthes and are separated from other metazoan organisms. 

A similar situation is found in the marine free-living and non-parasitic Placozoa, where two different but closely related channels were identified in *Trichoplax adhaerens* (~39% identity, ~54% similarity), with their genes separated by less than 10,000 bp (NW_002060945).

In Porifera, most species seem to have only one H_V_ channel. However, in the calcareous sponge *Sycon ciliatum*, three different H_V_ channels have been identified in TSA databases (see Appendix A). Future genomic and expression studies on Porifera are still needed as actual sequences analysis is somewhat scattered. 

In reef-building corals (Cnidaria) H_V_ channels were identified and characterized in *Acropora millepora* where they play an important role in calcification [124].

#### 2.1.4. H_V_ Channels in Lophotrochozoa

From the picture drawn so far within Metazoa, it could be concluded that most animal species have a single H_V_1 channel, some have none, and a few of them have two or even three separate genes. These changes are drastic, based on the analysis of the presence of putative H_V_ channel homologs in Lophotrochozoa. In this monophyletic superphylum, the number of putative H_V_ channels is utterly increasing, while the reasons for the presence of this large number of channels is still unknown. In Annelida (2), Entoprocta (3), Nemertea (3), and Bryozoa (3), Brachiopoda and Phoronidae (4), several gene duplications already increased H_V_ channel diversity within individual species (Figure 4). 

In the major Lophotrochozoan phylum, mollusks, up to eleven different H_V_ channels and their respective gene were found in a bivalvian species, the scallop *Mizuhopecten yessoensis*. The diversity is lower in the gastropode *Aplysia californica* (seven putative H_V_ channels) and in the cephalopode *Octopus bimaculoides* (four putative H_V_ channels) than in bivalvia. So far, four of these channels have been characterized: three in the gastropode *Aplysia californica* and one in the bilaterian *Crassostrea gigas* [125,126] (see below). Structurally, they can be divided into two groups: (1) conventional H_V_ channels (AcH_V_1, AcH_V_2, and CgH_V_4), characterized as rather small molecules (200–250 amino acids in length) with typical conserved residues for the selectivity filter in S1 (Asp64 in AcH_V_1) and voltage-sensor in S4 (RxWRxxR/K), and (2) unconventional H_V_ channels (AcH_V_3) with much larger intracellular domains, a very large extracellular S1–S2 loop and a slightly modified voltage-sensor motif (xPWRxxR). Database analysis showed that this later unconventional H_V_ channel family comprises a large proton channel family within the mollusk phylum, with huge variation in the length of the S1–S2 loop and intracellular domains and up to eight different members per species in bivalvia (manuscript in preparation). Outside the mollusk phylum, unconventional channels harboring a typical proline residue within S4 are found in a few species (Brachiopoda and Phoronida). In Annelida, Bryozoa, and Nemertea, H_V_ channels with large extracellular loops and large intracellular domains are present. However, the typical S4 proline residue of mollusk H_V_3 channels is missing. This leads to an evolutionary picture, where H_V_ gene duplication events and incorporation of large S1–S2 loops and intracellular domains took place in an early lophotrochozoan species, whereas the unusual proline residue entered the S4 segment only in a common ancestor of mollusks, brachiopodes and phoronides.

Speculatively, reasons why mollusks have so many different H_V_ channels might encompass: (1) As most mollusks live in a marine environment, a need for rapid adaptation to pH changes in their surrounding aqueous space may be of advantage. (2) The biomineralization process (calcification) during the formation of the exoskeleton in mollusks strongly depends on pH, and H_V_ channels might work in close conjunction with Ca^2+^-ATPases in this respect [127]. Indeed, even the calcification of mammalian otoconia is strongly dependent on a proton channel (Otopetrin-1) although this channel is structurally not related to H_V_ channels [128,129]. (3) A mollusk-specific function in the generation of action potentials maybe suggested as all three functionally analyzed Aplysia channels (AcHv1-3) are expressed in CNS/ganglia. A special need for H_V_ channel in mollusk may be caused by a higher metabolic rate in nerve fibers due to the lack of myelin sheath or even in a direct electrical participation of H_V_ channels in action potential formation. (4) As motifs for intracellular localization can be found within the sequence of several molluscan H_V_ channels, a putative function in the acidification of intracellular compartments was proposed. 

#### 2.1.5. H_V_ Channels in Fungi, Ichthyosporea, and Choanoflagelates

Beside Metazoa, H_V_ channels have been found in three other opisthokonta kingdoms: the fungi, where they also have been characterized electrophysiologically [130], in choanoflagelates, and in ichthyosporea. In all these kingdoms, only one H_V_ channel per species was found so far. Several fungi however lack voltage-gated proton channels, including yeast and *Neurospora crassa*. In identified fungal H_V_ channels, the consensus voltage-sensor motif (RxWRxxR) is somewhat modified as in these channels the third arginine is substituted by a lysine residue (RxWRxxK). The only channel found in ichtyosporea (*Amoebidium parasiticum*) has an interesting extension of the voltage-sensor motif reminiscent to H_V_ channels in protists, with a histidine residue three amino acids downstream the last arginine (RxWRxxRxxH) (Appendix A).

#### 2.1.6. H_V_ Channels in Plants

Proton channels are found in many red algae (Rhodophyta), green algae (Chlorophyta), and land plants (Embryophyta). As recently reviewed by Taylor et al. [131], these proton channels may have important functions in cellular pH homeostasis as well as in sensory biology, depending on the adaptation to different environments. Indeed, marine phytoplankton, freshwater algae, and land plants may have very different needs in their regulation of pH_i_ and H^+^ fluxes. Interestingly, proton channel genes are absent from many freshwater algae (e.g., *Chlamydomonas reinhardtii* or *Volvox*) and marine phytoplankton (*Ostreococcus*, *Micromonas*, and *Aureococcus*), whereas other members of the same order harbor typical H_V_ channels (see Appendix A; [131]). Whereas in red and green algae, only one H_V_ channel per species is found, at least some land plants, such as the spreading earthmoss *Physcomitrella patens,* may contain two separate genes (Appendix A).

#### 2.1.7. H_V_ Channels in Protists

Protists represent a large and diverse group of unicellular eukaryotes generally accepted to be at the origin of the evolution of eukaryotes. More than 1.8 billion years of eukaryotic evolution gave rise to a large diversity of H_V_ channels within protists. Whereas some protists (e.g., CRuMs) have no H_V_ channel homolog, some others have up to four different genes (*Stramenopiles*, see below). As a common sequence feature and in contrast to animal H_V_ channels, protist share the presence of a conserved histidine residue as an elongation of the voltage-sensor motif in S4 (RxWRxxRxxH) with most plants and Ichtyosporea. Only a few protist H_V_ channels have been described and characterized so far. In the coccolithophores *Emiliania huxleyi* and *Coccolithus pelagicus ssp braarudii* (Haptophyta), two H_V_ channels homologs were initially identified and physiologically characterized [132]. Their role in the process of calcification may be of general importance as ocean acidification is one of our main environmental problems to date. A database analysis showed that in *Emiliania* and two other members of the haptophyte order Isochrysidales two different proton channels exist (Appendix A). Similarly, the related Haptiste *Choanocystis* sp. (Centroplasthelida) possesses two distinct H_V_ channels. A bit unusual for protist H_V_ channels. The two *Choanocystis* H_V_s are a slightly larger and show N-terminal domains (>200 amino acids) and C-terminal domains (>150 amino acids) of significant length.

In dinoflagellates (Alveolata), H_V_ channels were characterized in *Karlodinium veneficum* [133] and in *Lingulodinium polyedrum* [134] where they are thought to trigger the bioluminescence flash (*Lingulodinium*). Subsequently, three major H_V_ subfamilies were identified in dinoflagellates, raising the possibility for additional proton channel functions beyond bioluminescence [135]. Database analysis identified H_V_ channel homologs also in Stramenopiles and Rhizaria (forming the supergroup SAR with Alveolata). Whereas in Rhizaria maximally two H_V_ channels were found per species (*Amorphochlora amoebiformis*, Appendix A) in at least one Stramenopiles, the marine diatom *Odontella aurita*, up to four different H_V_ channels and their respective genes may exist.

Finally, database analysis identified single protist H_V_ channels in Excavata (e.g., *Euglena gracilis*), Cryptista (e.g., *Chroomonas* sp.), and Amoebozoa/Eumycetozoa (e.g., *Raperostelium potamoides*) and up to two H_V_ channel homologs in a second Amoebozoa class (Discosea, *Balamuthia mandrillaris*). Although both *Balamuthia* H_V_’s share ~50% sequence identity within the S1–S4 core region, the different selectivity filter residue in S1 (Asp79 in GISS01003879 and Glu139 in GISS01013796) may indicate differing physiological properties and functions within these species. In Appendix A, a multiple sequence alignment combines the main H_V_ channels found in animals and protists.

#### 2.1.8. Summary 

In summary, H_V_ channels are widely expressed among eukaryotes, whereas no H_V_ channels are found in eubacteria and archaea. Many eukaryotes have at least one Hv channel gene. In several marine species, especially in species performing some form of biomineralization, such as mollusks, brachiopoda, stramenopiles, or haptophyta, the number of different H_V_ channels per species increases significantly. Bioluminescence, in alveolata, or the acidification of specialized intracellular compartments might be reasons for the existence of more than one Hv gene in some species. On the other hand, many species, including all modern insects (Diptera) and most of the nematodes, have no H_V_ channel, indicating that Hv channels are not a prerequisite for life.

### 2.2. Evolution beyond the Typical Proton Channels

Within the tree of life many gene duplication events gave rise to a huge amount of different proton channels with a vast variety of different structural features, such as different intracellular domains, extracellular loops of variable length, and characteristic point mutations within the selectivity filter motif or in the voltage-sensor. Most of the mutations should preserve proton channel function as long as the major H_V_ channel consensus sequences are conserved. However, it may not be surprising that in some species structures evolved from H_V_ channels have lost typical H_V_ channel consensus elements and therefore may not represent proton channels anymore. On the other hand, as they are expressed and do not represent *pseudo* genes, these channels may have gathered functions beyond just proton channels. In Appendix A, some examples are highlighted in red. A well-characterized example is the *Ciona intestinalis* voltage-sensor containing phosphates (Ci-VSP). Similar to H_V_ channels, the Ci-VSP has four transmembrane regions and a voltage-sensor in S4, The Ci-VSP sequence however misses a typical negatively charged residue in S1, serving as selectivity filter in H_V_ channels, and also has no tryptophan residue within the S4 segment. Not surprisingly, electrophysiological experiments showed gating currents underlying the movement of the voltage-sensor, but no typical proton selective current could be measured in WT Ci-VSP [136]. With the coupling of the Ci-VSP S1-S4 segment to an intracellular phosphatase domain, a new structural element arises combining functionality of both domains resulting in a voltage-gated phosphatase. At least one more structure with the same coupling of S1–S4 to a phosphatase was found in the ascidian tunicate *Styela clava* (XM_039409759).

Another example is the pelagic tunicate *Oikopleura dioica*. Here, two different but similar sequences (32% identity) where found, both harboring a five amino acid deletion within the S4 selectivity filter, including one of the voltage-sensing arginines and the conserved tryptophan residue (Appendix A). Some prediction algorithms do not recognize this shortened sequence element as a transmembrane region anymore, resulting in a protein with three transmembrane segments and an extracellular C-terminal domain (Appendix A).

In the hemichordate *Saccoglossus kowalevskii*, two sequences could be extracted from the GenBank database with the predicted definition “voltage-gated hydrogen channel” (see Appendix A). Indeed, both channels showed significant overall homology to H_V_ channels; however, one channel is a typical proton channel, and the second structure lacks a typical selectivity filter residue (Gln61 instead of Asp or Glu) as well as the consensus tryptophan in S4. Regarding these changes, a function as a proton-selective channel is unlikely.

Finally, also mollusks structures have evolved from the massive H_V_ gene duplications that may not resemble proton channels anymore. At first sight, even the members of the H_V_3 channel family were suspected to have lost proton channel function by their incorporation of large extracellular loops as S1–S2 linkers and their large intracellular domains. However, the functional expression of the *Aplysia* AcH_V_3 channel clearly showed that proton channel function is still preserved in these channels. On the other hand, some of the identified mollusk sequences have even lost further conserved sequence elements. For example, the sea snail *Limacina antarctica* has a S4 segment with voltage-sensing arginine residues and a S1 aspartate as selectivity filter; however, the H_V_-typical S4 tryptophan is missing. Future experiments will have to show if this “tryptophan-free” structure still retains its proton channel function since this residue has critical implications in H_V_ functioning [137,138].

## 3. Biophysical Properties of Functionally Tested H_V_ Channels among Species

The biophysical properties of H_V_ channels determine their physiological functions in different organisms. Although these channels exhibit common hallmark features such as perfect proton selectivity, pH- and voltage-regulated H^+^ conduction, strong temperature-dependent H^+^ permeation, inhibition by divalent cations (e.g., Zn^2+^), among others, there are subtle yet important differences intrinsic to the function of each channel. In this section, we focus on twenty proton channels characterized electrophysiologically from various groups of organisms and describe three main biophysical features: selectivity, voltage-dependent gating, and pH-dependent gating. The compiled data for the selected H_V_ channels is summarized in Table 4.

### 3.1. Proton Selectivity

The selectivity of H_V_ channels for protons is perhaps the most striking feature of these molecules and proved to be very high. The selectivity of a voltage-activated ion channel is described by the Goldman–Hodgkin–Katz (GHK) equation (Equation (1)). Accordingly, the ion with the higher concentration and/or greater permeability is likely to dominate the electrochemical equilibrium or reversal potential (*V*_rev_).

(1)
$$V\mathrm{rev}=\frac{RT}{F}\;\ln\left(\frac{\sum_{i}^{n}P_{M_{i}^{+}}{\left[M_{i}^{+}\right]}_{out\;}+\sum_{j}^{m}P_{A_{j}^{-}}{\left[A_{j}^{-}\right]}_{in\;}}{\sum_{i}^{n}P_{M_{i}^{+}}{\left[M_{i}^{+}\right]}_{in\;}+\sum_{j}^{m}P_{A_{j}^{-}}{\left[A_{j}^{-}\right]}_{out\;}}\right)$$

*V*_rev_ = reversal potential (in V or J ∙ C^−1^); *R* = ideal gas constant (in J ∙ mol^−1^ ∙ K^−1^); *T* = temperature (in K); *F* = Faraday constant (in C ∙ mol^−1^); [*M*^+^]*_out_* = extracellular concentration of the cation *M*^+^ (in M); [*M*^+^]*_in_* = intracellular concentration of the cation *M*^+^ (in M); *P_M_*^+^ = permeability for the cation *M*^+^ (m ∙ s^−1^); [*A*^−^]*_out_* = extracellular concentration of the anion *A*^−^ (in M); [*A*^−^]*_in_* = intracellular concentration of the anion *A*^−^ (M); *P_A_*^−^ = permeability for the anion *A*^−^ (in m ∙ s^−1^).

(2)
$$E_{H}=\frac{RT}{zF}\;\ln\frac{{\left[H^{+}\right]}_{out}}{{\left[H^{+}\right]}_{in}}$$

*E*_H_ = Nernst potential for protons (in V); *z* = ion charge (1^+^); [*H*^+^]*_out_* = extracellular concentration of protons (in M); [*H*^+^]*_in_* = intracellular concentration of protons (in M).

The selectivity of an H_V_ channel can then be tested by measuring the reversal potential experimentally, at different proton gradients, and comparing with the Nernst potential for protons, *E*_H_ (Equation (2)). Figure 5 illustrates an example of the experimental determination of the selectivity of a H_V_ channel using electrophysiology. 

Deviations of *V*_rev_ from *E*_H_ values are associated to poor experimental control of the pH_i_, which can occur when there is depletion of the protonated buffer [86]. Hence, in small cells, such as those used as expression systems in patch-clamp studies, proton depletion is common at large depolarization. Protons conducted through H_V_ channels leave the cell faster than the buffer can be replenished, due to the diffusion time from the pipette to the cell and to the dissociation time in the cytoplasmic bulk solution. Proton depletion leads to a rise in pH_i_ and a consequent shift in *V*_rev_ from its nominal value (*E*_H_) to more positive potentials. Conversely, negative deviations of *V*_rev_ from *E*_H_ may occur in rare cases where H_V_ channels permit inward H^+^ conduction that acidifies the cytosol, making the accurate determination of *V*_rev_ challenging. 

The direction of H^+^ flux is determined by both the pH- and voltage-dependence of gating, which is a characteristic of each channel. Inward directed conduction is possible at large inward pH gradients (pH_o_ < pH_i_) for most of H_V_ channels, even those considered as proton extruders. However, for a few H_V_ channels, H^+^ influx occurs even at symmetrical pH conditions. For instance, the channel of the insect *Extatosoma tiaratum* (EtH_V_1) exhibits robust inward H^+^ currents during depolarization and slow closing times after hyperpolarization, leading to the acidification of the internal milieu that rapidly changes *V*_rev_ during electrophysiological protocols [122]. Furthermore, the aberrant type-3 H_V_ channel from the mollusk *Aplysia Californica* (AcH_V_3) leaks H^+^ selective currents in the closed-state, constantly acidifying the cytoplasm and inducing new equilibriums at different voltage-pulses [125].

Although there are some experimental limitations in evaluating the selectivity of a proton channel, all H_V_ channels tested to date have demonstrated perfect selectivity regardless of their ionic environment. The minimal H^+^/ion permeability rates are given in Table 4. The values depicted are either as reported by the authors or calculated from the experimental conditions as the quotient between the minimal proton concentration (higher pH) and the ion present in higher concentration. 

Under physiological conditions, the concentration of protons is <10^6^ times lower than that of other ions, e.g. the intracellular concentration of potassium ([K^+^]_i_) is around 150 mM, whereas the intracellular concentration of protons ([H^+^]_i_) is approximately 100 nM. Given this, the selectivity mechanism of proton channels must be considered extremely efficient.

By comparing the amino acid sequences of H_V_ channels to the voltage-sensor domains (VSD) of other voltage-activated cation channels and of the non-conducting but voltage-sensing protein C15orf27 (TMEM266) [150], Musset et al. identified five amino acid residues potentially responsible for the exclusive proton conduction in H_V_ channels. Single mutation of each of those candidates in the human proton channel (hH_V_1) to those found at equivalent positions in C15orf27, permitted to identified Asp112 as the selectivity filter of H_V_ channels [142]. Furthermore, substitution of this aspartate to non-acidic amino acids made hH_V_1 permeable to anions, an observation that was also confirmed in dinoflagellate and insect H_V_ channels [121,133].

Asp112 is located directly above a constriction of the channel that separates the internal and external solutions, approximately in the middle of the S1 alpha helix [143,151,152], and is highly conserved among H_V_ molecules. To date, the only tested H_V_ channel that presents a naturally occurring variation at this position is EtH_V_1. EtH_V_1 possesses a glutamate residue as a selectivity filter instead of the typical aspartate, a feature common to H_V_ channels from stick and gladiator insects [122]. Despite this peculiarity, EtH_V_1 elicited H^+^ currents that were consistent with *E*_H_ along different pH gradients (ΔpH = pH_o_ − pH_i_), as it was observed by experimentally mutating the selectivity filter Asp to Glu in channels from other species [121,133,142].

Multiple evidence as the high temperature dependence of H_V_ [47,88], the diminished deuterium conductance [83], the H^+^ hopping between pairs of charged residues lining H_V_ pore detected in QM/MM molecular dynamics simulations [153], or more recently the proposed existence of an interrupted water wire within H_V_1 pore [154], suggest that a protonable carboxyl group is essential for the selectivity of H_V_ channels. However, the mechanism underlying this selectivity involves also specific physicochemical microenvironments where H^+^ conduction and selectivity are coupled [121,133,142,155,156,157,158,159]. Thus, previous studies have shown that moving Asp112 along S1 helix of the channel affects proton selectivity [157], and further quantum mechanical calculations have revealed that an Asp can interact with a counter charge carried by the Arg sidechain to create an electrostatic barrier that repels other ions besides H^+^ [158]. The selectivity is, nevertheless, maintained when Arg is replaced with Lys [158]. While some studies support the interaction between Asp112 and Arg208 in S4 [121,143,151,157,160], others propose an alternative explanation where Asp112 interacts with Arg211 instead [159,161,162,163]. A 3D structure of hH_V_1 showing Asp112 in S1, and R208 and R2011 in S4, is displayed in Figure 2 (left). 

According to the analysis of a vast collection of mutant H_V_ channels, DeCoursey et al. proposed the minimal requirements of the selectivity filter of H_V_ channels: a carboxyl group (Asp or Glu) facing the pore in a narrow region of the channel and the ability to interact with a counter charge (Arg or Lys) [164].

### 3.2. Molecule Architecture and the Voltage-Dependent Gating

Voltage-gated proton channels can detect electrochemical changes across the cell membrane and consequently open or close their conduction pathway; a process called gating. With membrane depolarization, H_V_ channels open their gates and generate time-dependent H^+^ currents, while membrane hyperpolarization leads to channel closure. H_V_ channels are formed by a bundle of four membrane-spanning alpha helices that are structurally similar to the VSD of other voltage-sensitive ion channels such as K_V_ and Na_V_ channels (Figure 2). Despite its resemblance to other cation channels, the constitutive pore segments, S5 and S6, are absent in H_V_ channels. Proton channels are VSDs that selectively conduct H^+^ [111,119].

Experiments aimed at determining the stoichiometry of the channel have confirmed its dimeric nature across several species, including mammals, insects, ascidians, sea urchins, and reef-building corals [41,123,124,139,140,144,146,148,160] (Table 4). H_V_ subunits are held together by a coiled-coil structure at the cytosolic C-terminal [139,140,144,160,165]. The only possible exceptions to H_V_ oligomerization come from the unicellular organisms *Karlodinium veneficum* and *Phaeodactylum tricornutum*, where informatic tools did not predict a coiled-coil region at the C-terminal, thus enabling a potential monomer expression [133,166]. Dimerization is important for the regulation of other properties, such as channel kinetics or the inhibition by divalent cations, e.g., H_V_ monomers activate faster and are less sensitive to Zn^2+^ than dimers [167]. Moreover, H_V_ gating is also modulated by the cooperativity between the channel subunits [146,168,169,170,171]. Despite the importance of oligomerization for the function of H_V_ channels, when the C-terminal is truncated to force the expression of monomeric forms, the resulting subunits demonstrate their own functional pore, conserving properties such as selectivity and voltage- and pH-dependence of gating [139,144,155]. 

Depolarization induces conformational changes in H_V_ that precede proton conduction and drive the channel from a closed to an open configuration. It is commonly agreed that the S4 alpha helix moves upward before H_V_ conduction, as proved by voltage-clamp fluorometry and patch clamp recordings [162,169,172,173]. Other studies suggest that other parts of the channel, such as S1, may also move during the transition from closed to open configuration [162]. Furthermore, these conformational changes are regulated not only by the membrane potential but also by the ΔpH [81,174,175,176,177] (Figure 6).

In order to provide a comprehensive understanding of H_V_ gating, numerous mathematical models have been proposed. Cherny et al. generated the first model in 1995, for the rat channel RnH_V_1, which described gating regulation by pH with a three-state model [81]. Patch clamp measurements of dimeric and monomeric H_V_s suggested multiple voltage-dependent conformational changes confirming the Cole–Moore effect [97,147]. The voltage-dependent gating of *Ciona intestinalis* proton channel (CiH_V_1) was described by Gonzalez et al. using also a three-state model, with each transition translocating two effective gating charges [173]. In CiH_V_1 monomers, Carmona et al. measured gating currents and proposed that a model describing H_V_ activation should have more than two-state kinetic steps (≥5 states) [147]. Analyzing the kinetics of human hH_V_1, Villalba-Galea proposed that the activation and the deactivation processes have different transitions, and suggested that a voltage-independent transition occurs before H_V_ opening [178]. Chaves et al. produced a simplified model based on two-state kinetics for the Nicoletia channel (NpH_V_1) that described the channel activation kinetic well using parameters representing a dimer (power [*n* = 1.5–2.1]), similar to the ones used by Fujiwara et al. (power [*n* = 1.7]) in their kinetics analysis of the mouse mH_V_1 [148,179]. More recently, Suárez-Delgado et al. used a non-natural fluorescent amino acid set on the top of S4 of hH_V_1 and suggested the possibility of further movements in the VSD, even after channel opening. The authors generated a four-state model for channel activation similar to the voltage-dependent kinetic proposed by Villalba-Galea in 2014 [175]. Rangel-Yescas et al. conducted studies on the reef-building coral AmH_V_1 and created an allosteric eight-state model that couples proton binding to opening transitions [124]. 

Upon careful examination of the gating mechanism of proton channels, it becomes clear that the process is complex and lacks consensus among researchers. However, it is widely accepted that the voltage-sensing of H_V_ channels works similarly to the other canonical VSDs [180,181,182]. In these VSDs, positive charges located at the S4 segment move across a narrow hydrophobic region of the VSD that separates the external and internal milieu. This hydrophobic region focuses the electric field across the cell membrane to form the charge transfer center (CTC) and it can be delimited by a conserved phenylalanine residue in S2 [182,183,184,185] (Figure 2). 

Similarly, H_V_ channels have a conserved Phe residue in S2 (Phe150 in hH_V_1) that belongs to a ring of hydrophobic residues located approximately in the middle of the membrane, called hydrophobic gasket (HG) [186] or hydrophobic plug [187]. The hydrophobic gasket regulates H_V_1 activation [188], and consequently, H_V_ activation can be defined by the relative movement of charged amino acids across a point demarcated by amino acids at positions equivalent to Phe150 (Figure 6). In the S4 segment, H_V_ channels possesses a series of three positively-charged arginine residues (R1, R2, and R3) organized in an archetypal RxWRxxR voltage-sensor motif (VSM) that are responsible for voltage sensing [173]. This characteristic VSM motif is highly conserved among most H_V_ channels and, together with the selectivity filter in S1, has been used to identify H_V_ homology sequences in TSA databases [121,122,124,125,126,133,148].

The movement of charged residues through the CTC during H_V_ gating can be studied by determining the elementary charges (*e*_0_) that theoretically may correspond to arginine residues crossing the entire electrical field [189] or negatively charged residues moving in the other direction. In this regard, the estimation of effective gating charges in H_V_ channels has been achieved by different methods, such as the limiting slope method of the *G–V* relationship in semi-log plots [148,173,190], Boltzmann fittings of normalized *G–V* curves [111,146], or direct measurement of gating currents [141,147]. As seen in Table 4, estimations in different species harboring H_V_ channels show elementary charges ranging from 1.1 to 2.7 *e*_0_ (monomer) and 4.3 to 6.1 *e*_0_ (dimer), corresponding to three arginines in S4 per monomer. 

The function of each charge carrier in voltage-sensing and H_V_-functioning channel has been assessed separately. Neutralization of the first arginine (R1) in hH_V_1 (R205A mutant) (*zδ* = 0.57, *V*_0.5_ = 99.5, ΔpH = −1) reduced the apparent voltage-dependence of gating compared to the WT channel (*zδ* = 0.90, *V*_0.5_ = 58.0, ΔpH = −1), indicating a drop in the effective charge moved [111]. Mutation of R1 to glutamine accelerates the activation kinetics and shifts the *g*_H_–*V* relationship to more negative potentials, enabling the mouse H_V_1 to activate at more negative values than *V*_rev_, thus allowing H^+^ influx [123]. In hH_V_1, R1 interacts with Asp185 (in S3) during gating, suggesting a change in the residue’s position during the process [191]. R1 is well conserved among H_V_ channels with the exception of some unconventional channels, as the type-3 H_V_ of *A. californica* (AcH_V_3) that displays a natural substitution to a hydrophobic leucine to generate an atypical LPWRxxR VSM. Interestingly, AcH_V_3 voltage-dependent gating appears to be altered, with reduced steepness of *g*_H_–*V* curves (~60 mV/*e*-fold change) compared to AcH_V_1 and AcH_V_2 channels (4–6 mV/*e*-fold change), which in contrast retain the RxWRxxR VSM typical of regular H_V_ channels [125]. The reduced voltage-dependence of gating of AcH_V_3 is hypothesized to be related to the leak of protons in the channel’s closed state and/or the inability of being measured at very negative potentials. Similarly, alterations of the physicochemical properties of the hydrophobic gasket (HG) of hH_V_1 generate H^+^ leak in the closed state [186].

Notably, the second arginine (R2) of the VSM is fully conserved in all H_V_ channels found so far. R2 appears to have a dual role in H_V_ channels, both acting as a charge carrier and interacting with Asp112 in the conducting state [177], to putatively become an essential component of the proton-selectivity mechanism [158]. In hH_V_1, arginine to histidine mutants were tested with Zn^2+^ in patch clamp experiments to reveal state-dependent interactions. During the essays, in the open state, both R2H and R1H were accessible to externally applied Zn^2+^. In the closed state, R2H mutant was internally accessible but not R1H, indicating a journey of R2 from the inner to the external water vestibule during H_V_ activation [151]. Jardin et al., 2022, [177] suggested that the outward S4 movement focuses on the central region of the VSM, where R2 shuttles between the inner and outer vestibules of hH_V_1. Wu et al., 2022, proposed that Phe150 in the HG interacts with R2 in the closed state but with R3 in the open state [188].

Water accessibility studies have provided evidence that the third arginine (R3) is exposed to the intracellular medium in the closed configuration [143,151,154,157,160,173,177]. However, controversy arises regarding the journey of R3 and its final position in the open configuration. Different research groups have suggested that R3 interacts with Asp112 in the open state configuration, which supports a *two-click* hypothesis in *Ciona* and human H_V_s [159,161,162,163]. In contrast, other studies support a *one-click* hypothesis [151,158,160,177]. Neutralization of R3 to alanine in NpH_V_1 did not impair H^+^ selectivity at pH_o_ = pH_i_ = 5.5 [121]. R3K charge-conserved mutations in hH_V_1 conducted by Wu et al. revealed that the mutation affected H^+^ conduction by slowing the activation kinetics and shifting the channel conductance to more positive potentials. The study showed that R3–Phe150 interaction is important for the stabilization of the open state, as R2 moves further upwards during the closed to open transition [188]. Moreover, truncation of mH_V_1 between R2 and R3 did not prevent H_V_ voltage-dependence [160]. 

The conservation of R3 among all H_V_ channels is not fully established, and natural variations have been recently discovered. For instance, fungal H_V_ channels and the newly discovered CgH_V_4 from the pacific oyster display a natural variation to a lysine (K3) at positions equivalent to R3, which generates a RxWRxxK signature [126,130]. This naturally occurring K3 is conserved in all H_V_4 channels of mollusks and in fungal H_V_1. The only exception is found in the fungi *P. brasilianum*, which has a glutamine variation. Gating charge calculations obtained from *g*_H_–*V* relationship slopes resulted in ~5 *e*_0_ for AoH_V_1 and SlH_V_1 fungal channels, consistent with the ~6 *e*_0_ characteristic of dimeric H_V_s from other species (Table 4). There is no report of gating charge calculations or CgH_V_4 stoichiometry in Chaves et al., 2023b. However, CgH_V_4, similar to AoH_V_1 and SlH_V_1, displays a lag phase in the onset of H^+^ currents that may indicate cooperativity between H_V_ subunits [146,167]. The exposition of R3 to the intracellular solution would translate to p*K*_a_ values for both R3 and K3 to be close to those reported for bulk solution (>10). Therefore, it is likely that K3 is protonated at physiological conditions and effectively serves as a gating charge carrier. The fact that a lysine residue can act as an effective gating charge carrier in VSDs is uncommon, as there is a peculiar prevalence of arginines over lysines. The reason for such a phenomenon is still not completely understood [192].

### 3.3. The pH-Dependent Gating and Its Physiological Implications

The modulation of the proton conductance (*g*_H_) by pH is a ubiquitous feature of H_V_ channels and is fundamental to their physiological function. For most of the cases and particularly in mammals, the channel functions like an overpressure valve that opens as the concentration of H^+^ in the interior of the cell rises, thereby regulating cellular pH_i_ homeostasis.

The pH dependence of gating of H_V_ channels has been commonly described by the *rule of forty*, established by Cherny et al. for rat RnH_V_1 in 1995 [81], and apply to native channels in other species [193]. This rule states that H_V_ channels shift their *g*_H_ to more positive or negative potentials in ~40 mV per unit pH. The modulation of *g*_H_ is independent of the side of the membrane where the pH change is applied (pH_i_ or pH_o_) [81]. Therefore, a cellular process that acidifies the cytosol by a unit of pH (decreasing pH_i_) or alkalizes the extracellular space (increasing pH_o_) by a unit of pH will set the proton conductance to a membrane potential 40 mV more negative than its nominal value. Since most of native H_V_ channels open ~20 mV positively to *E*_H_ at symmetrical pH [98,193], the *rule of forty* explains why H_V_ channels are considered proton extruders. 

The amino acid residues responsible of ΔpH sensing remain a mystery. Ramsey et al., 2010, extensively studied single mutations of ionizable residues in hH_V_1, but none of the neutralized residues affected substantially the Δ40 mV/pH unit sensitivity [156], potentially pointing to the involvement of multiple groups of amino acids [181]. Moreover, the underlying molecular mechanism is independent of the characteristic position of the *g*_H_–*V* curve along the voltage-axis of each channel. For example, *the rule* applies for both the *Danio rerio* channel (DrH_V_1) and *Mus musculus* channel (mH_V_1), although the first one activates more negatively than the latter [145]. 

Nevertheless, a few exceptions to the *rule of forty* have been reported in the last years (Figure 7, bottom-right). For instance, the proton channel from the snail *Helisoma trivolvis* HtH_V_1 displays an anomalous ΔpH-dependent gating with weak pH_i_ sensitivity (~15 mV/unit pH_i_) while its pH_o_ sensitivity is increased (~60 mV/unit pH_o_) [149]. Other molluscan H_V_ channels also present alterations in ΔpH-dependent gating. In *A. Californica*, AcH_V_1 and AcH_V_2 channels have identical pH_o_ dependence of gating, which still falls within the range of the *rule of forty* (~43–45 mV/pH_o_ unit). Yet, AcH_V_1 has a weak pH_i_ sensitivity similar to that of HtH_V_1, while AcH_V_2 pH_i_ sensitivity is normal (~40 mV/pH_i_ unit) [125]. *V*_0.5_ vs. ΔpH plots, displayed by the channel from the reef-building coral *Acropora millepora* (AmH_V_1), show a slope steeper than 40 mV/ΔpH that deviates from linearity to saturate at high ΔpH [124]. Remarkably, fungal AoH_V_1 and SlH_V_1 exhibit the greatest deviation from the rule of forty (80–90 mV/ΔpH unit). Both channels have similar ΔpH-dependent gating, but SlH_V_1 displays an abnormal pH_i_ sensitivity within the pH range from 5.5 to 6.5 (~18 mV/pH_i_ unit), whereas for the same pH range, conduction of AoH_V_1 appears to be regulated exclusively by ΔpH instead of the absolute pH [130]. The NpH_V_1 and EtH_V_1 insect channels, the molluscan CgH_V_4, and the dinoflagellate kH_V_1 have all greater pH-dependent gating ranging between 45–50 mV/ΔpH [121,122,126,133]. There is also loss of linearity in the Δ*g*_H_–*V*/ΔpH relationship at certain pH values in some species. For instance, the *g*_H_–*V* curves of LpH_V_1 show saturation at pH_o_ > 8 and pH_i_ > 7.0 [134]. Similarly, the pH-dependence of gating of mammalian hH_V_1 and RnH_V_1, the coccolithophore channel from EhH_V_1, and the dinoflagellate kH_V_1 show deviations at pH_o_ > 8.0 [137]. In HtH_V_1, there is no sign of saturation of *g*_H_–*V* curves, but the slope of the *V*_gH,max/10_/ΔpH relationship increases at larger ΔpH [149]. The deviations from the *rule of forty* are hypothetically related to the strength of allosteric coupling between the voltage sensor and the proton-binding sites [124], while the saturation of *g*_H_–*V* curves at large ΔpH are thought to be indicative of the pk_a_ values of those proton binding sites [81].

A molecular mechanism coupling the pH- and voltage-sensing would be of great help to understand the pH-dependent gating of H_V_ channels. Despite this mechanism is yet unknown, several mathematical and atomistic models have been proposed to describe it. Markin established the first mathematical model for RnH_V_1 [81], and recently, Rangel-Yescas et al. developed a model for the reef coral building [124]. However, both models lack a clear explanation regarding how protonation of internal and external sites affect H_V_ gating. In Sokolov et al. [194], two molecular mechanisms are proposed to regulate simultaneously: an electrostatic mechanism and a *countercharge* model. In the electrostatic mechanism, protons binding to the channel affect the way H_V_ sense voltage, while in the *countercharge* model, protonation of acidic residues within the S1–S3 segments destabilizes salt bridge interactions that promote H_V_ active or inactive states. Ramsey et al. [156] proposed that the pH gradient and the voltage-dependence of activation are regulated by the interaction of protonated waters with the S4 arginine residues in the pore of the channel. Jardin et al., 2022 [177] used a constant-pH molecular dynamics approach to obtain the protonation states of putative ionizable residues and the conformational changes in hH_V_1 driven solely by the influence of the pH. Similarly, Carmona et al. [176] suggested that the ΔpH-dependent gating of CiH_V_1 is a consequence of the motions of the voltage sensor movements in a state-dependent manner, regulated by pH. Using patch-clamp fluorometry and proton uncaging, Schladt and Berger proposed that ΔpH provokes changes in the VSM in S4 that couple voltage and pH sensing in CiH_V_1 [174]. Han et al. [195] also mentioned the affectation of pH on S4 movement of hH_V_1. The authors concluded that from three different conformations acquired by S4, two in the deactivated (intracellular and middle) and one in the activated (extracellular) configuration, solely the conformational transition between deactivated-intracellular to activated-extracellular displays both voltage- and pH-dependence. To date, the only residue that has been identified to alter pH sensing significantly is His168 (in hH_V_1 nomenclature) [196]. This His residue, located at the bottom of S3, close to the S2-S3 linker, and below Phe150, is critical for pH_i_ sensing. The residue also accounts for the slow activation kinetics displayed by mammalian H_V_ channels, as other faster channels from other species, such as *H. trivolvis*, *L. stagnalis* and *S. purpuratus*, show natural variations to Gln or Ser [196].

The pH-dependent gating of H_V_ channels can be more easily described by a linear equation of the form:*V*_thres_ = a·*V*_rev_ + b(3)

a = slope (*V*_thres_/*V*_rev_) and b = *V*_offset_ (mV)

Thus, the pH dependence of a proton channel can be characterized by measuring the threshold voltage of activation (*V*_thres_) and the reversal potentials (*V*_rev_) in a wide range of ΔpH. While *V*_thres_ is an arbitrary parameter (the conductance of a channel is determined by the open probability as function of voltage), it can be generally estimated during patch-clamp protocols as the most negative voltage where the onset of H^+^ currents are observed, typically identified by the appearance of a first distinctive tail current after depolarizing pulses [98].

In order to compare the selected H_V_ channels, we used reports from studies that have evaluated the pH-dependence of gating according to Equation (3). In cases where this information was not directly provided, we used *V*_thres_ at symmetrical pH conditions close to neutrality (e.g., pH 6.5–7.0), either explicitly stated by the authors or deduced from *g*_H_–*V* relationships at different ΔpH, to obtain the linear function. The modulation of *g*_H_ by pH as described before is restricted to certain pH ranges though. Figure 7 (bottom, left) displays the results for a ΔpH_o_ range from −1 to +1 for 18 H_V_ channels based on the information compiled in Table 4. The dotted line in Figure 7 shows parity between *V*_rev_ and *V*_thres_. A channel activation negative to *E*_H_ (*V*_thres_ < *V*_rev_, data fitting below the dotted line) implies a driving force moving protons inwardly, and vice versa. In the absence of a pH gradient (ΔpH = 0), the direction of H^+^ flux is readily described by *V*_offset_ in Equation (3). This method permits a quick visualization of the direction of the H^+^ flux for the different channels.

The H^+^ flux direction, in conjunction with the physicochemical environment of the channel in physiological conditions, might indicate the role of H_V_ channels in the different species. For example, the type-1 and type-2 H_V_ channels from the mollusk *A.californica* have identical voltage-dependent gating (same *V*_thres_/*V*_rev_ slope) but differ markedly in *V*_offset_ (Figure 7, upper right). While AcH_V_1 is a regular proton extruder, AcH_V_2 displays vigorous H^+^ inward currents to presumably generate action potentials related to the respiratory pumping reflex of the animal [125]. The AoH_V_1 from the fungus *Aspergillus oryzae* opens, when the electrochemical H^+^ gradient is inward, to consequently acidify and depolarize the cell. Nevertheless, its huge voltage-dependent gating (~90 mV pH/unit) would make the channel to act as a H^+^ extruder under suitable pH conditions [130]. Likewise, the abnormal ΔpH-dependent gating of HtH_V_1 might theoretically allow H^+^ influx under specific conditions [149]. The dinoflagellate LpH_V_1 displays a very positive *V*_offset_ (+46 mV), theoretically pointing to a proton extrusion function. However, LpH_V_1 carries inward H^+^ currents to trigger bioluminescence, as a consequence of large proton gradients found in *Lingulodinium* vacuoles (ΔpH = 3.5) [134]. In contrast, the other dinoflagellate channel, kH_V_1, has a significantly negative *V*_offset_ (−37 mV) and shows robust inward H^+^ currents even at symmetrical pH conditions. *Karlodinium* kH_V_1 conducts inward H^+^ currents over a wide pH-range, whose function is presumably related to feeding, as *Karlodinium* is not bioluminescent [133]. The channel from the insect *Extatosoma* (EtH_V_1) also presents robust H^+^ influx which may be related to different functions, including neuronal depolarization, regulation of ventilation rates, or the control of the acid–base transport systems [122]. Curiously, the other insect channel, NpH_V_1, displays pH-dependent gating identical to EtH_V_1, in the same voltage range, but its more positive *V*_offset_ makes NpH_V_1 a regular proton extruder [122].

### 3.4. Summary: Biophyscial Properties of H_V_ Channels

The selectivity for protons in H_V_ channels is a distinctive characteristic. Experimental determination involves comparing the reversal potential with the Nernst potential for protons, although deviations can occur due to poor control of intracellular proton concentration. Despite experimental limitations, H_V_ channels have demonstrated an exceptional proton selectivity of at least 10^6^ P_H+_/P_other ion_, indicating a highly efficient mechanism. Specific residues, particularly Asp112, play a critical role in proton selectivity, and mutations of this residue can alter the channel permeability. Neutralizing Asp112 renders H_V_ channels from humans, dinoflagellates, and insects permeable to anions. Asp112 is conserved and, in a proposed mechanism, interacts via salt bridge with a counter charge arginine near the channel constriction, establishing the selectivity filter. To date, the only variation of Asp112 is found in some polyneopteran insects, where a glutamate residue replaces it. Nevertheless, the charge is conserved and proton selectivity in such H_V_ channels remains unaffected.

H_V_ channels undergo gating in response to changes in membrane potential. They consist of four membrane-spanning alpha helices similar to the voltage-sensing domain (VSD) of other voltage-sensitive ion channels but lack specific pore segments. In nature, H_V_ channels exist as dimers, and this oligomerization is important for regulating various properties. The gating mechanism of H_V_ channels is complex and involves conformational changes in response to depolarization and/or pH differences. Several mathematical models have been proposed to describe H_V_ gating, with the consensus being that the movement of the S4 segment is crucial. Three arginine residues in the S4 segment (R1, R2, and R3), organized in a typical RxWRxxR voltage-sensing motif (VSM), play a key role in voltage sensing and are thought to serve as the charge carriers. These charged residues move through a hydrophobic region in the middle of the channel (delimited by a conserved Phe residue) and their individual contribution has been studied. The first charge, R1, is exposed to the extracellular solution during gating and is naturally substituted by a lysine residue only in H_V_3 channels. H_V_3 channels have an atypical LPWRxxR VSM and display poor voltage sensing and H^+^ leakage in the closed state. R2 is fully conserved in all species and interacts with Asp112 in the open state to form the selectivity filter. Nevertheless, R3 is also proposed to interact with Asp112 in the open state, and there are different opinions regarding its final position in the open configuration. R3 is exposed to the intracellular solution in the closed configuration, and natural substitutions to lysine (K3) have been discovered recently in molluskan H_V_4 and fungal H_V_ channels.

The modulation of proton conductance (*g*_H_) by pH in H_V_ channels determines their physiological significance. This regulation has been commonly described by the “rule of forty” and applies for most H_V_ channels. The property states that the channel shifts *g*_H_ by approximately 40 mV per unit pH change, independently of the side of the membrane where the pH change occurs. However, several exceptions to this rule have been recently reported in different species, indicating variations in pH sensing. Some channels exhibit deviations from linearity or *g*_H/_ΔpH saturation at certain pH values. These deviations are thought to be related to the strength of allosteric coupling between the voltage sensor and proton-binding sites as well as an indication of the p*k*_a_ values of these binding sites. The molecular mechanism coupling pH and voltage sensing is not yet fully understood, and the amino acid residues responsible for pH sensing remain unknown. Various mathematical and atomistic models have been proposed, but none provide a clear explanation of how protonation of internal and external sites affects H_V_ gating. Some models suggest that proton binding affects voltage sensing or destabilizes salt bridge interactions involved in channel activation. By using a linear relationship involving the threshold voltage of activation (*V*_thres_) and reversal potentials (*V*_rev_), the pH-dependent gating of H_V_ channels can be compared and evaluated. Moreover, the H^+^ flux direction can be determined, and in conjunction with the physicochemical environment of the channels, insights into the functional roles of H_V_ channels in different species can be gained. For example, most H_V_ channels act as proton extruders, particularly in mammals, where their functioning is important to avoid the accumulation of acid in the cytosol or extreme depolarization of the cell membrane. In contrast, other species such as mollusks, insects, fungi, and dinoflagellates have H_V_ channels that allow H^+^ influx under specific conditions. In these species, H_V_ functions may range from triggering action potentials to initiating bioluminescent flashes.

## Figures and Tables

**Figure 1 biomolecules-13-01035-f001:**
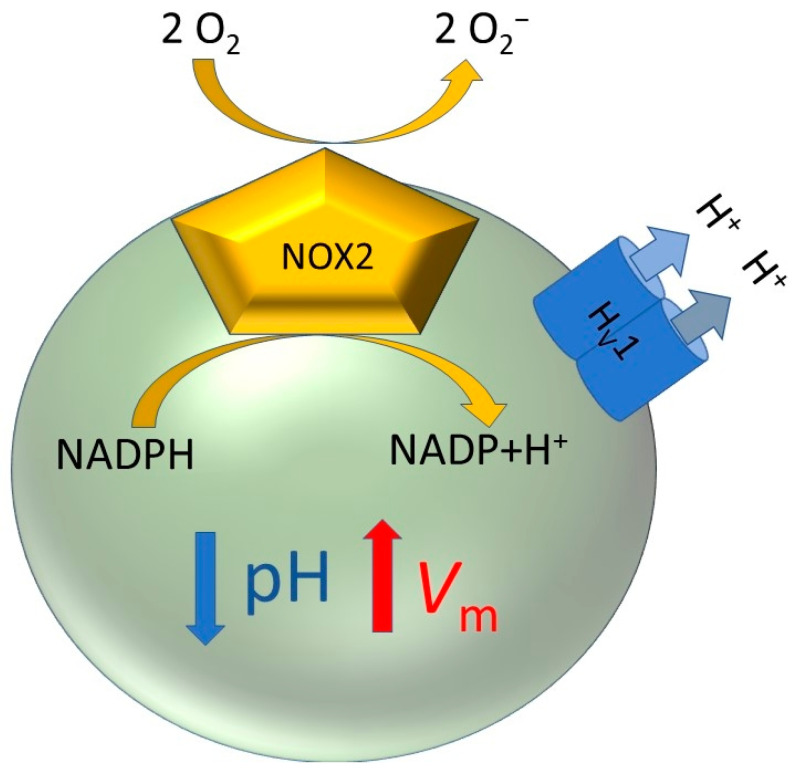
Simplified scheme of the respiratory burst in phagocytes. The activity of the NADPH oxidase reduces intracellular pH and depolarizes the plasma membrane. The voltage-gated proton channel H_V_1 compensates for the charge by conducting protons out of the cell. Additionally, the outflux of protons counteracts the decreased pH. Therefore, H_V_1 and NADPH oxidase are perfect synergistic partners during the respiratory burst.

**Figure 2 biomolecules-13-01035-f002:**
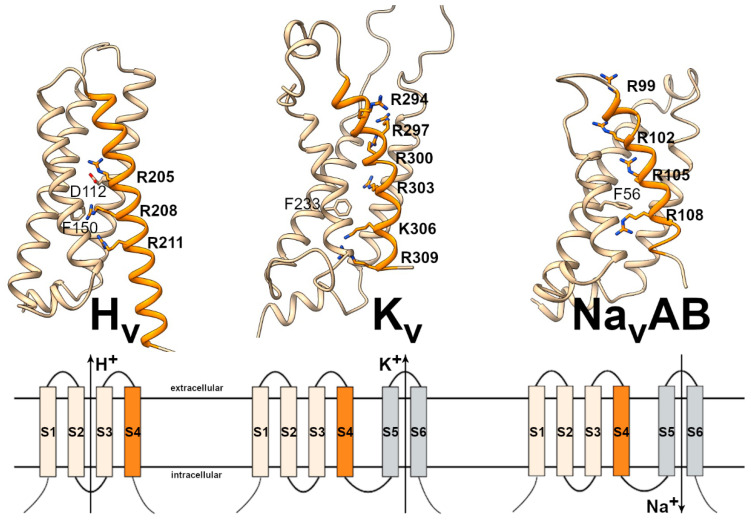
**Architecture of the voltage-sensing and conduction pore domains of H_V_, K_V_, and Na_V_ channels.** As examples, a homology model of the human H_V_1, the crystal structure of the K_V_1.2 (PDB: 3LUT [109]), and the bacterial Na_V_AB channels (PDB 3RVY [110]) are shown. The voltage-sensing domain of all voltage-gated channels consists of four transmembrane helices (S1–S4). In H_V_ channels, the voltage-sensing domain is also the ion conduction pore. In K_V_, Na_V_, and also Ca_V_ channels, the ion conduction pore is formed by a tetrameric assembly of two further transmembrane helices S5 and S6. The voltage-sensing arginines, three in H_V_, four in Na_V_, and five plus a lysine in K_V_ are shown as sticks. Upward, resp. downward movement of the S4 (orange) helix is coupled to the activation, resp. deactivation, of the channels upon membrane depolarization, resp. hyperpolarization. A phenylalanine, shown as sticks, in the hydrophobic gasket that separates the intracellular and extracellular vestibules of the channels is conserved among voltage-gated ion channels.

**Figure 3 biomolecules-13-01035-f003:**
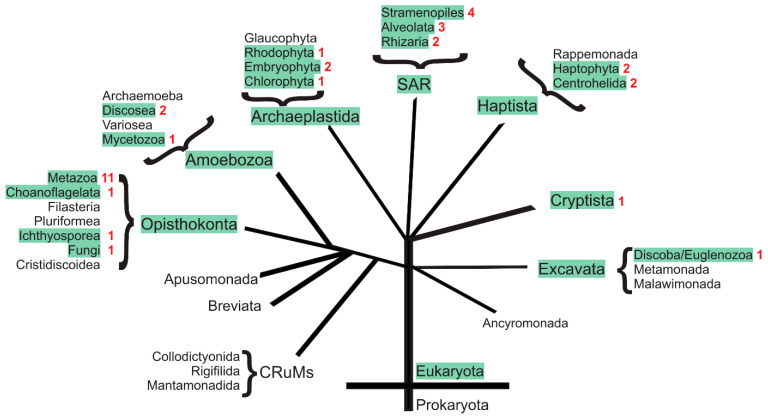
**H_V_ channels within the eukaryotic tree of life.** Phylae and Subphylae containing at least one species harboring an H_V_ channel gene are marked in green. The maximal number of different H_V_ channel genes per species is indicated in red.

**Figure 4 biomolecules-13-01035-f004:**
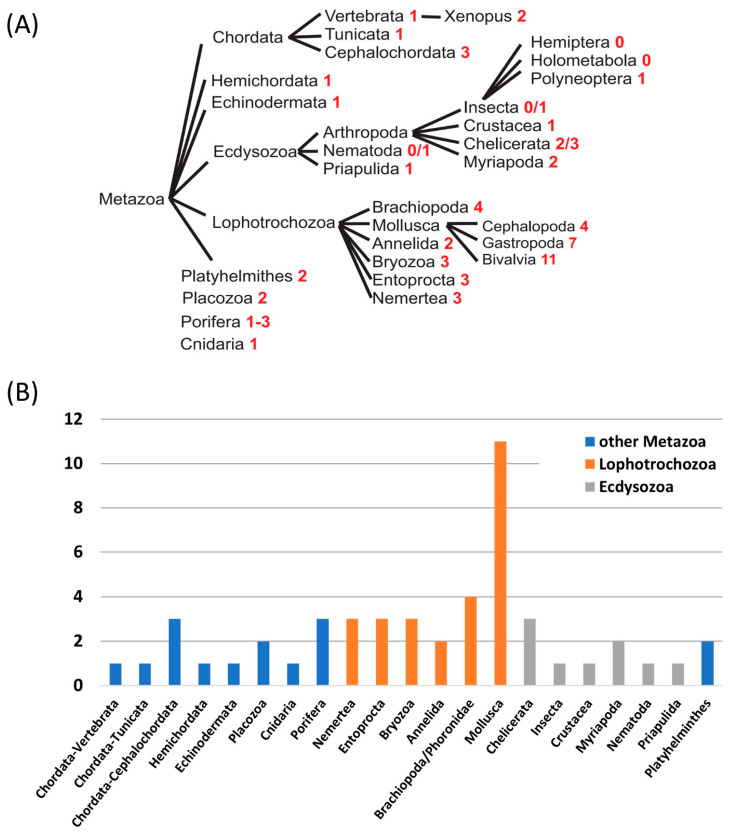
**H_v_ channels in Metazoa.** (**A**) Tree of metazoan life. The maximal number of different H_V_ channel genes per species is indicated in red. (**B**) Maximal number of H_V_ channels genes per species in different phylae.

**Figure 5 biomolecules-13-01035-f005:**
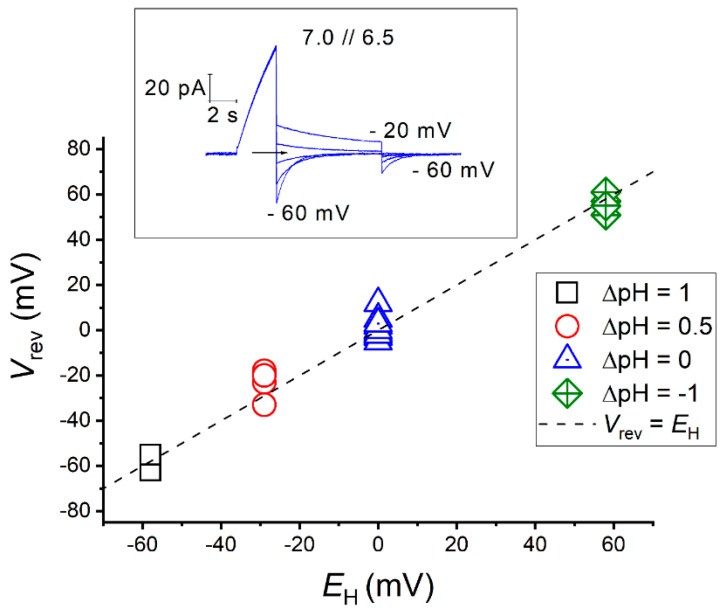
Single *V*_rev_ measurements of the H_V_4 channel from *Crassostrea gigas* are plotted against the *E*_H_ predicted value at different proton gradients (ΔpH = pH_o_ – pH_i_). The dotted line shows equality between *V*_rev_ and *E*_H_. The **inset** depicts an example of a *V*_rev_ experimental determination by the tail currents method in a whole-cell patch clamp configuration at ΔpH = 0.5. Repolarization of the cell membrane after a depolarizing pulse was conducted in Δ10 mV/step from −60 mV to −20 mV) with a holding potential of −60 mV. The arrow indicates the point where H^+^ currents reverse (*V*_rev_), here ~33 mV. Taken from [126].

**Figure 6 biomolecules-13-01035-f006:**
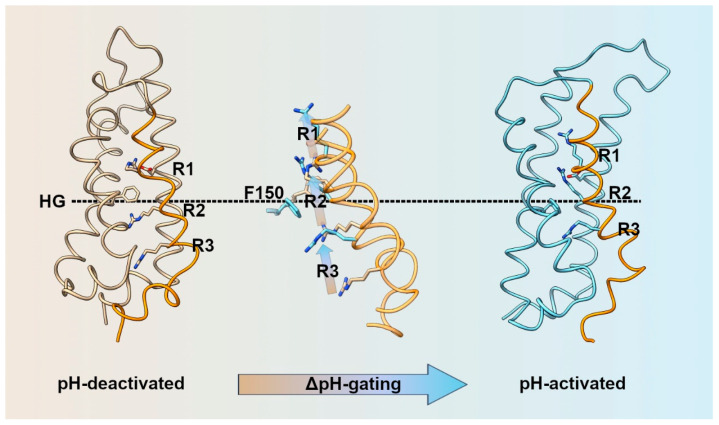
**ΔpH-gating of the human H_V_1 channel from molecular dynamics simulations**. The ΔpH-dependent gating from a deactivated (left, brown) to an activated (right, blue) conformation moves the three gating charges R1 to R3 in S4 (orange ribbon) outwards, passing the second arginine R2 through the hydrophobic gasket (HG). Changes of the location and orientation of the gating charges in the two conformations result in different interaction networks both in the internal and in the external vestibules, as illustrated for the selectivity filter aspartate (D112 in hH_V_1, here in red) that interact with R1 in the deactivated but with R2 in the activated conformation. For more details, please see [177].

**Figure 7 biomolecules-13-01035-f007:**
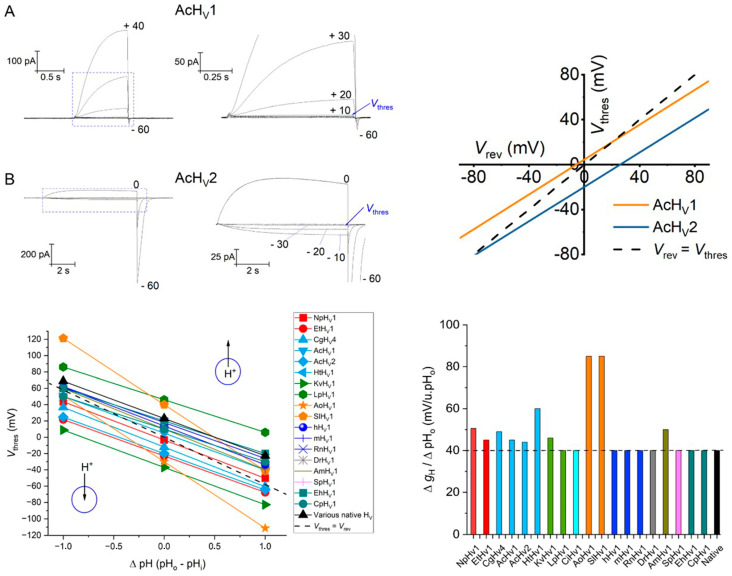
**pH-dependent gating of H_V_ channels in different organisms.** The evaluation of the pH-dependent gating of H_V_ channels can be described by a linear function of the form *V*_thres_ = *slope*·*V*_rev_ + *offset* from data obtained at different ΔpH conditions. (**Upper panel, left**) Families of H^+^ currents elicited by two different proton channels from *Aplysia californica* at symmetrical pH = 6.5. A *zoom-in* of dashed areas for each family is shown on the right side. The membrane potential where currents are first detected (*V*_thres_) differs for both channels. AcH_V_1 activates positive to *V*_rev_ (here at +10 mV) and H^+^ currents are then outwardly directed (**A**). In contrast, AcH_V_2 activates negative to *V*_rev_ (here at −30 mV), permitting H^+^ influx (**B**). (**Upper panel, right**) pH-dependent gating of AcH_V_1 (orange line) and AcH_V_2 (blue line) represented according to Equation (3). Both channels present identical pH-sensing with the same slope (*V*_thres_/*V*_rev_) but have distinct offsets. The dotted line represents equality between *V*_thres_ and *V*_rev_ and delimits the equilibrium between outward and inward driving forces. Data over and below the dotted line indicate outwardly and inwardly directed H^+^ driving force, respectively. While AcH_V_1 extrudes H^+^ alkalinizing the cytosol, AcH_V_2 conducts inward currents that acidify the cell. From [125]. (**Lower panel, left**) *V*_thres_ vs. ΔpH (pH_o_ − pH_i_) plot determined for some functionally tested H_V_ channels from different species in a ΔpH ranging from −1 to +1. The pH-dependent gating of native H_V_ channels of 15 cell types (from distinct organisms including rat, hamster, mouse, human, frog, and snail) described by [193] is displayed for comparison (black solid triangles). The dotted line shows equality between *V*_rev_ and *V*_thres_. Most of H_V_ channels are proton extruders but there are few notable exceptions, i.e., the channels from the dinoflagellate *K. veneficum* (kH_V_1), the fungi *A. oryzae* (AoH_V_1), the stick insect *E. tiaratum* (EtH_V_1), or the type-2 channel from sea hare *A. californica* (AcH_V_2). (**Lower panel, right**) the pH-dependent gating on the external pH_o_ for several H_V_ channels is represented as Δ*g*_H_/ΔpH. The values indicate the shift of the conductance–voltage curves expected once pH_o_ is changed in one unit while maintaining pH_i_ constant. The dotted line shows the 40 mV/unit of pH thumb rule describing the pH-dependent gating for most of H_V_ channels [193]. The channels are grouped by organism in insects (red), mollusks (light blue), dinoflagellates (green), ascidians (light cyan), fungi (orange), mammals (blue), fishes (grey), corals (dark green), sea urchins (pink), coccolithophores (dark cyan), and 15 native H_V_ from different species. Np: *Nicoletia phytophila*, Et: *Extatosoma tiaratum*, Cg: *Crassostrea gigas*, Ac: *Aplysia californica*, Ht: *Helisoma trivolvis*, Kv: *Karlodinium veneficum*, Lp: *Lingulodinium polyedrum*, Ci: *Ciona intestinalis*, Ao: *Aspergillus oryzae*, Sl: *Suillus luteus*, h: *homo sapiens*, m: *Mus musculus*, Rn: *Rattus norvegicus*, Dr: *Danio rerio*, Am: *Acropora millepora*, Sp: *Strongylocentrotus purpuratus*, Eh: *Emiliania huxleyi*, Cp: *Coccolithus pelagicus ssp braarudii*.

**Table 1 biomolecules-13-01035-t001:** H_V_1 in human tissue and other mammalian tissue.

Respiratory Burst	Cell Type	Function of H_V_1	References
**Yes**	Eosinophil (human) (rodent)	Charge compensation, prevention of cell death.	[10,12,17,36,37,38,39,40,41,42,43,44,45,46]
**Yes**	Neutrophil (human) PLB-985 (hcl) HL-60 (hcl) K-562 (hcl) Neutrophil (rodent)	Charge compensation, migration, granula release, calcium homeostasis, pH homeostasis, ERK activity, phagosomal pH homeostasis.	[6,8,10,12,13,14,15,41,45,47,48,49,50,51,52,53]
**Yes**	Monocyte (human)	Charge compensation.	[54]
**Yes, small**	Macrophage (human) THP-1 (hcl) Macrophage (mice)	Charge compensation, phagosome acidification.	[52,55,56]
**Yes, small**	Osteoclast (rodent/Leporidae)	pH homeostasis, charge compensation, ROS production.	[57,58,59,60,61]
**Yes, small**	Microglia (rodent) Microglia culture (human) BV-2 (rcl) GM1-R1 (rcl) MLS-9 (rcl)	pH homeostasis, charge compensation, ROS production, microglia-astrocyte communication, neuropathic pain promotion, brain damage enhancing, acidosis exacerbation, M2 polarization reduction, demyelination promotion, white matter injuries promotion, secondary spinal cord damage enhancing, neuroinflammation promotion, pyroptosis increase, motor deficit expansion, autophagy increase, M1 polarization promotion in aged mice.	[22,61,62,63,64,65,66,67,68,69,70,71,72,73,74,75,76,77]
**Yes**	Kupffer cell (mice/rodent)	Glucose metabolism, ROS production suppression, hyperglycaemia, and hyperinsulinemia prevention.	[23]
**No**	Cardiac fibroblast (human)	pH homeostasis, membrane potential, potentially beneficial in ischemia.	[78]
**Yes, small**	Dendritic cells (rodent/human)	TLR9 activation.	[28]
**No**	Sperm cell (human)	Capacitation, acid extrusion.	[32,33,79,80]
**Yes**	Oocyte (human)	pH homeostasis.	[34]
**No**	Type 2 alveolar cells (rodent)	pH regulation.	[5,30,47,81,82,83,84,85,86,87]
**No**	Mast cell (mouse)	pH homeostasis.	[88]
**Yes, tiny**	B cells (human) (rodent) LK35.2 (rodent)	B cell receptor signalling, migration and proliferation enhancing (short isoform).	[26,27,89,90]
**No**	T cells (human) Jurkat (human) T cells (rodent)	Apoptosis prevention, pH homeostasis, autoimmune disorders prevention.	[29,89,91,92,93]
**No**	Cardiomyocytes (canine)	pH homeostasis.	[11]
**No**	SHG-44 glioma cells (human)	Apoptosis prevention.	[18]
**No**	Colorectal cancer (human) SW620 (hcl) HT29 (hcl) LS174T (hcl) Colo205 (hcl)	Prevention of cellular acidosis, support of cancer cell metabolism, pH homeostasis, potential biomarker, and drug target.	[19]
**No**	Basophils (human)	Exocytosis (histamine release), pH homeostasis.	[16,94]
**No**	Ovary cells (Hamster)	pH homeostasis.	[95]
**No**	Breast cancer cells (human primary) MDA-BA-231(hcl) MCF-7 (hcl) MDA-MB-468 (hcl) MDA-MB-453 (hcl) T-47D (hcl) SK-BR-3 (hcl)	Tumor growth, metastasis and invasiveness promotion, (expression predicts prognosis of tumor).	[20,21,96]
**No**	Lung cancer cell A549 (human)	No information.	[97]
**No**	Prostate cancer cell PC-3 (human)	No information.	[97]
**No**	Kidney (human) HEK-293	No information.	[97,98]
**Yes, small**	Nasal epithelium (human primary culture) JME/CF 15 (human) Cystic fibrosis genotype	Airway surface epithelium acidification, proton extrusion.	[99]
**Yes, small**	Ciliated tracheal cells (human)	NADPH oxidase activity driven proton extrusion.	[31,99]
**Yes, small**	lung epithelium fetal (human)	DUOX driven proton release, acid extrusion.	[100]
**Yes, tiny**	Serous gland cell line Calu-3 (human)	Airway surface epithelium acidification, proton extrusion (to a lesser extent than airway epithelium).	[31]
**No**	Skeletal muscle myocyte (human)	pH homeostasis.	[7]
**No**	Glioblastoma cell line (human) T98G	Cell’s survival and migration.	[101]
**No**	Whole heart (rodent)	NOXs transcription and CO_2_ homeostasis control, electrophysiological remodelling.	[102]
**No/Yes**	Vascular system, Immune system	Atherosclerosis advancement (hypothetical).	[103]
**No/Yes Whole tissue**	Lung (rodent)	Goblet cell hyperplasia prevention. Depression expression of IL-4, IL-5, and IL-13. Reduction of the expression levels of NOX2, NOX4, and DUOX1. Promotion of the expression of SOD2 and catalase. Reduction of the development of allergic asthma through ROS production enhancing.	[104]
**Yes**	Myeloid derived suppressor cells (MDSC) (rodent)	T-cells regulation (via ROS production).	[35]
**No**	epididymal adipose tissue (rodent)	Diet obesity induction.	[24]
**Yes, tiny**	Pancreatic β cells (rodent)	Insulin secretion, ROS production, NOX4 upregulation, glucotoxicity induction.	[25,105,106]

hcl = human cell line; rcl = rodent cell line.

**Table 2 biomolecules-13-01035-t002:** **Sequence homology in H_V_ channels.** Sequence identities/similarities of ten selected protist H_V_ channels to eukaryotic sodium and calcium channels are shown. In brackets, the length of the respective sequence as retrieved by BLAST is shown.

Species	Phylae/Subphylae	Na_V_	Ca_V_
*Euglena gracilis*	Excavata	28/52% (86)	35/50% (91)
*Raperostelium potamoides*	Amoebozoa	36/51% (106)	35/56% (86)
*Balamuthia mandrillaris*	Amoebozoa	34/52% (74)	<25%
*Paramoeba aestuarina*	Amoebozoa	27/50% (64)	<25%
*Emiliania huxleyi*	Haptista	37/51% (79)	32/57% (73)
*Amorphochlora amoebiformis*	SAR-Rhizaria	<25%	28/55% (79)
*Odontella aurita*	SAR-Stramenopiles	36/58% (67) *	<25%
*Karlodinium veneficum*	SAR-Alveolata	<25%	34/55% (80)
*Scrippsiella hangoei*	SAR-Alveolata	29/51% (87)	29/56% (85)
*Gracilaria vermiculophylla*	Rhodophyta	30/63% (69)	<25%

* In one case to a prokaryotic sodium channel.

**Table 3 biomolecules-13-01035-t003:** **Identified H_V_ channels.** GenBank accession numbers of cDNA and genes of indicated model organisms are listed.

Species	Kingdom	cDNA (mRNA)	Gene
*Escherichia coli*	Prokaryota	-	-
*Dictyostelium discoideum*	Protist	-	-
*Tetrahymena thermophila*		-	-
*Naegleria gruberi*		-	-
*Emiliania huxleyi*		HBNU01018021 GIZZ01010784	NW_005196830 NW_005202428
*Thalassiosira pseudonana*			NC_012076
*Aspergillus niger*	Fungi	XM_001390088	NT_166520
*Neurospora crassa*		-	-
*Saccharomyces cerevisiae*		-	-
*Arabidopsis thaliana*	Plantae	NP_001321473	NC_003070
*Zea mays*		-	-
*Oryza sativa*		-	-
*Physcomitrella patens*		XM_024508236 XM_024525718	NC_037254 NC_037259
*Marchantia polymorpha*		-	AP019868 AP019873 AP019871
*Chlamydomonas reinhardtii*	Archaeplastida	-	-
*Aplysia californica*	Invertebrata	XM_005100609 XM_005093050 XM_005094218 XM_013086351 XM_013080089 XM_013090418 XM_013082371	NW_004797523 NW_004797327 NW_004797348 NW_004797727 NW_004797344 NW_004798539 NW_004797441
*Branchiostoma belcheri*		XR_002139895 XM_019760615 XM_019764911	NW_017802379 ” NW_017803191
*Caenorhabditis elegans*		-	-
*Ciona intestinalis*		NM_001078469	NW_004190496
*Daphnia pulex*		XM_046604416	NC_060027
*Drosophila melanogaster*		-	-
*Hydra vulgaris*		XM_047284425	-
*Lymnaea stagnalis*		FX197150 FX190227 FX196339	nys
*Nematostella vectensis*		XP_001626501	NC_064038
*Strongylocentrotus purpuratus*		XM_030990962	NW_022145605
*Trichoplax adhaerens*		XM_002110878 XM_002110360	NW_002060945 ”
*Ambystoma mexicanum*	Vertebrata	GFZP01114012	JXRH01463164
*Gallus gallus*		NM_001396354	NC_052587
*Mesocricetus auratus*		XM_040731183	NW_024429206
*Cavia porcellus*		XM_003462980	NT_176304
*Mus musculus*		NM_028752	NC_000071
*Rattus norvegicus*		XM_017598517	NC_051347
*Macaca mulatta*		XM_028829869	NC_041764
*Takifugu rubripes*		XM_003977031	NC_042298
*Xenopus laevis*		XM_018249100 XM_018244209	NC_054371 NC_054372
*Danio rerio*		NM_001002346	NC_007121
*Homo sapiens*		NM_001040107	NC_000012

nys = not yet sequenced.

**Table 4 biomolecules-13-01035-t004:** Biophysical properties of some H_V_ channels.

Organism	Species	Channel	Oligomerization?	Selectivity	Gating Charges, *e*_0_	Slope *V*_thres_/*V*_rev_	*V*_thres_ at ΔpH = 0 (mV)	Δ*g*_H_-*V*/ΔpH (mV/pH_o_)	H^+^ Influx at Relevant Physiological pH?	References
Mammals	*H. sapiens*	hH_V_1	confirmed ^f,g,j,k^	>10^6^ P_H+_/P_TMA+_ ^e,i^ >10^6^ P_H+_/P_CH3SO3-_ ^i^ >10^6^ P_H+_/P_Cl-_ ^i^	~5 ^h,δ^ ~6 ^l^	0.82 ^e^ 0.67–0.71 (expressed) ^l^ 0.71 (native) ^l^	13.8 ^e^ −9 to −11 (expressed) ^l^ +27 (native) ^l^	40 ^l^	no (native) yes (if expressed) ^l^	[111] ^e^, [139] ^f^, [140] ^g^, [141] ^h^, [142] ^i^, [41] ^j^, [143] ^k^, [98] ^l^
	*M. musculus*	mH_V_1	confirmed ^n^	>10^7^ P_H+_/P_NMDG+_ >10^7^ P_H+_/P_Na+_ >10^7^ P_H+_/P_K+_	~6 ^m^	0.86 * (expressed) 0.69 ^m^ (expressed)	+10 to +20 −15 (expressed) ^m^	50 40 ^m^	no (native) yes (if expressed) ^m^	[119], [98] ^m^, [144] ^n^
	*R. norvegicus*	RnH_V_1	possibly	>10^7^ P_H+_/P_TMA+_ ^o^ >10^8^ P_D+_/P_TMA+_ ^p^	5.4 ^p^	0.76	+18	44 40 ^o,p^	no	[5], [81] ^o^, [83] ^p^
Fish	*D. rerio*	DrH_V_1	possibly	>10^7^ P_H+_/P_NMDG+_	n.d.	0.69 *	~+10 mV ^ε^	~40 ^ε^	no	[145]
Sea squirt	*C. intestinalis*	CiH_V_1	confirmed ^c^	n.d.	4.4–5.9 (dimer) ^c^ 1.6–2.7 (monomer) ^d^	n.d.	n.d.	~40 ^d^	no	[119], [146] ^c^, [147] ^d^
Insects	*N. phytophila*	NpH_V_1	confirmed ^b^	>10^8^ P_H+_/P_TMA+_ >10^4^ P_H+_/P_Na+_ >10^4^ P_H+_/P_Cl-_	4.7–6.1 ^b^	0.81 ^a^	−3.4 ^a^	47–54 ^a^	no	[121] ^a^, [148] ^b^
	*E. tiaratum*	EtH_V_1	n.d.	>10^6^ P_H+_/P_TMA+_	n.d.	0.77	−23	45	yes	[122]
Mollusks	*C. gigas*	CgH_V_4	possibly	>10^7^ P_H+_/P_TMA+_	n.d.	0.84	−12	49	no ^●^	[126]
	*A. californica*	AcH_V_1	possibly	>10^7^ P_H+_/P_TMA+_ >10^6^ P_H+_/P_Na+_ >10^6^ P_H+_/P_K+_	5.7	0.78	5	43–45	no	[125]
	*A. californica*	AcH_V_2	possibly	>10^7^ P_H+_/P_TMA+_ >10^6^ P_H+_/P_K+_	5.3	0.77	−20	44 40 (pH_i_)	yes	[125]
	*A. californica*	AcH_V_3	possibly ^+^	>10^7^ P_H+_/P_TMA+_	n.d.	n.d.	n.d.	n.d.	yes ^§^	[125]
	*H. trivolvis*	HtH_V_1	possibly	>10^7^ P_H+_/P_TMA+_	5.5	1.03 * 0.26 (pH_i_) *	n.d.	60.0 15.3 (pH_i_)	no	[149]
Corals	*A. millepora*	AmH_V_1	confirmed	>10^7^ P_H+_/P_TMA+_	2 ^λ^	0.86 *	~+10 mV ^θ^	~50 ^θ^	no	[124]
Sea Urchin	*S. purpuratus*	SpH_V_1	confirmed	>10^7^ P_H+_/P_K+_	4.3 (dimer) 1.1 (monomer)	0.69 *	~+10 mV	~40 ^β^	no	[123]
Fungi	*A. oryzae*	AoH_V_1	possibly ^+^	>10^5^ P_H+_/P_TEA+_ ^#^	5	1.40–1.55 *	~−30 (pH 5.5) ^γ^ ~−30 (pH 6.5) ^γ^	80–90	yes	[130]
	*S. luteus*	SlH_V_1	possibly ^+^	>10^5^ P_H+_/P_TEA+_ ^#^	5	1.40–1.55 *	~+20 (pH 5.5) ^γ^ ~+40 (pH 6.5) ^γ^	80–90	no	[130]
Dinoflagellates	*K. veneficum*	kH_V_1	possibly not ^φ^	>10^7^ P_H+_/P_TMA+_ >10^5^ P_H+_/P_Cl-_	n.d.	0.79	−37	46	yes	[133]
	*L. polyedrum*	LpH_V_1	possibly ^+^	>10^9^ P_H+_/P_TMA+_	n.d.	0.69 *	46	40 **	yes ^α^	[134]
Phytoplankton	*E. huxleyi*	EhH_V_1	possibly	>10^6^ P_H+_/P_K+_ >10^6^ P_H+_/P_Cl-_	n.d.	0.69 ^μ^ (expressed)	~+20 mV (expressed) ^μ^	~40 ^Ω^	no	[132]
	*C. pelagicus*	CpH_V_1	possibly	>10^6^ P_H+_/P_K+_ >10^6^ P_H+_/P_Cl-_	n.d.	0.69 ^μ^ (native)	+10 mV (native) ^μ^	~40 ^Ω^	no	[132]

* Calculated from conductance shifts (∆*g*_H_-*V*/∆pH) and *E*_H_ = 58 mV·pH^−1^. ^§^ AcH_V_3 leaks protons at the closed state [125]. ^#^ Calculated from pH = 6.0 and working solution containing 30 mM of TEA^+^ as main cation. ^●^ CgH_V_4 has an activation of −12 mV which permits proton influx at symmetrical pH (physiological) conditions. Nevertheless, the proximity of the offset value to *V*_rev_ makes the inward H^+^ electrochemical gradient small. Thus, H^+^ influx is small at ∆pH = 0 where currents rectify rapidly with depolarization, behaving more like a typical H_V_ proton extruder. During measurements, strong and/or consistent H^+^ influx currents were not detected. ^φ^ kH_V_1 lacks the predicted coiled-coil which is important for H_V_ dimerization, a potential monomeric expression is discussed by the authors [133]. ^+^ Protein sequence analysis predicts a C-terminal coiled-coil domain. ** LpH_V_1 pH-dependent gating saturates above pH_o_ 8.0 [134] similar to rat, human, *Karlodinium*, and *Emiliania* H_V_ channels [137]. ^α^ Inward H^+^ currents are onset at large inward pH gradients (1–3 ∆pH units) [134]. ^β^ estimated from *g*_H_-*V* curves shift between pH_o_ 6.5 and 7.0 (~20 mV), at pH_i_ = 7.0 (Figure 2; [123]). The *g*_H_-*V* shifts between pH_o_ 6.0 to 6.5 and 6.5 to 7.0 are not identical due to experimental limitations reported by the authors. The calculated value on pH_o_-dependent gating agrees with Δ*g*_H_-*V*/pH_i_ unit (Appendix A; [123]). ^γ^ from normalized *g*_H_-*V* curves (Figure 3; [130]). ^δ^ from Q_on_ of dimeric W207A-N214R mutant ([141]). ^θ^ from *V*_thres_-ΔpH and slope of *V*_0.5_-ΔpH curves (Figure 6; [124]). ^λ^ apparent from steepness of normalized *g*_H_-*V* (Figure 6; [124]). ^Ω^ from current densities (Figure 1; [132]). ^μ^ the slope *V*_thres_/*V*_rev_ was calculated from shifts of *I*_H_ onsets and *E*_H_. *V*_offset_ was estimated shifts of *I*_H_ activation, *E*_H_, and Equation (3) (Figure 4; [132]). P_H+_ = proton permeability, P_TMA+_ = tetramethylammonium permeability, P_TEA+_ = tetraethylammonium permeability, P_K+_ = potassium permeability, P_Na+_ = sodium permeability, P_Cl-_ = chloride permeability, P_NMDG+_ = N-methyl-d-glucamine permeability, P_CH3SO3-_= methanesulfonate permeability, n.d. = no data, *V*_thres_ = threshold potential, *V*_rev_ = reversal potential, ∆pH = proton gradient (pH_outside_ – pH_inside_), *g*_H_-*V* = proton conductance – voltage relationship, pH_o_ = external pH, pH_i_ = internal pH.

## Data Availability

All data is included in the manuscript or Appendix A.

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
