# Peer review of "Voltage-Gated Proton Channels in the Tree of Life"

_biomolecules, 2023, doi:10.3390/biom13071035_

Round 1
Reviewer 1 Report
The review by Chaves et al addresses "Voltage-gated proton channels in the tree of life". It is divided into three parts, starting by discussing a new role of HV-channels in mammals in close functional interaction with NADPH oxidases. In the second part, the evolutionary tree of Hv-channels is presented, while in the third part, biophysical properties and molecular dynamics of Hv-channels are discussed. the authors did a great job to collect all the data, but as a result , the review is very dense with information and thus difficult to digest. It is aimed more at experts in the field of voltage-gated channels who have a high interest in details. The reader who wants to get an overview of Hv channels can easily get lost in this thicket of information and data. It would be very helpful if there were a short summary of the most important information at the end of each block, especially sections 2 and 3. An alternative would be to split the review and present either sections 1 and 3 or 2 and 3 together. Then it would also be possible to make section 1 a bit more detailed. What might be the role of the Hv channel in cells that do not show an oxidative burst?
It would be interesting to combine the data on the different channel forms in the different species and geneara with speculation/ respective data on the special demands, respectively environmental/physiological conditions for the species/genera that have evolved in parallel. For example, can you speculate what is the specific need of mollusks to have this abundance of Hv-channels? Also, the appearance of different Hv channels in Plants seems rather random?
Spefic comments:
Section 1.2 is almost without references, these are found in table 1. I would suggest to integrate references in the text
Table 4: explain abbreviations
the structural model is presented very late, my suggestion is to move it before section 2
Since the biophysical mechanisms of action are discussed in such detail from page 21 onwards, it would be helpful to have molecular models for the different active states to illustrate what is written in the large paragraph from page 21 onwards.
Reviewer 2 Report
The manuscript by Chaves and colleagues reviews voltage-gated ion channels from an evolutionary perspective. The manuscript is informative, particularly the tables and figures that summarise various findings discussed throughout the text. I anticipate that the data collected in the manuscript will provide valuable references for the broad readership interested in voltage-gated ion channels. I thus recommend publication.
That being said, I have found the manuscript somewhat superficially written. Lots of superfluous wording such that, when reading the manuscript, one largely lacks the sense of carefully crafted text. Evaluations (personal?) such as 'Hv1 is the perfect partner of the NADPH oxidase', or 'The system is then free to run', or 'Fortunately, as described in the next chapter', 'The situation for ... is a bit different', 'proved to be flawless', are just some examples of wording that give the impression of superficial scientific writing.
'do/does/may not/neither posses', 3 words (lines 182, 312 335, 380, 400) expresses the same as 'lack(s)', 1 word.
Reviewer 3 Report
Chaves et al. wrote a very comprehensive review about voltage-gated proton channels. The review is well written, thoroughly organized, and easy to read. It covers the functional
expression of Hv channels in mammals and its role in respiratory burst of phagocytes, and in independent functions such as pH homeostasis or acid extrusion. The review also covers the evolution of Hv channels and the immense diversity of proton channels in other phylae compared to only one signle gene in mammals. It ends with an overview of the biophysical properties of the different HV channels characterized so far.
I do not have any concerns, but I have few suggestions which will complete the comprehensiveness of this review. The hypothesis that proton channel is evolved by being truncated, S1-S4 domain, from other voltage gated ion channels is supported by the recent discovery of gating pore current (Sokolov et al, 2005, Neuron). It is resulted from missense mutations of the gating charges in S4, which make the VSD either permeable to protons or cations. Adding a section which elaborate more on this idea would be beneficial to readers. In the same respect, the structure of the gating pore current in a bacterial sodium channel was resolved in 2018 (Jiang D., Gamal El-Din T.M. et al., 2018, Nature). It should be mentioned and used in this discussion.
I would like to thank the authors for this very well written review.
- Minor editing of English language required
Round 2
Reviewer 1 Report
the manuscript has improved by adding summaries and Figures, it is a real comprehensive review